# Which vaccine attributes foster vaccine uptake? A cross-country conjoint experiment

**Sabrina Stöckli**[1]* , **Anna Katharina Spälti**[1], **Joseph Phillips**[2], **Florian Stoeckel**[1], **Matthew Barnfield**[1]* , **Jack Thompson**[1], **Benjamin Lyons**[3], **Vittorio Mérola**[4], **Paula Szewach**[1], **Jason Reifler**[1]

**1** Department of Politics, University of Exeter, Exeter, United Kingdom, **2** School of Psychology, University of Kent, Canterbury, United Kingdom, **3** Department of Communication, University of Utah, Salt Lake City, Utah, United States of America, **4** Department of Political Science, Stony Brook University, Stony Brook, NY, United States of America

☯ These authors contributed equally to this work.

* s.stoeckli@exeter.ac.uk (SS); m.barnfield@exeter.ac.uk (MB)

## Abstract

Why do people prefer one particular COVID-19 vaccine over another? We conducted a pre-registered conjoint experiment (n = 5,432) in France, Germany, and Sweden in which respondents rated the favorability of and chose between pairs of hypothetical COVID-19 vaccines. Differences in effectiveness and the prevalence of side-effects had the largest effects on vaccine preferences. Factors with smaller effects include country of origin (respondents are less favorable to vaccines of Chinese and Russian origin), and vaccine technology (respondents exhibited a small preference for hypothetical mRNA vaccines). The general public also exhibits sensitivity to additional factors (e.g. how expensive the vaccines are). Our data show that vaccine attributes are more important for vaccine preferences among those with higher vaccine favorability and higher risk tolerance. In our conjoint design, vaccine attributes–including effectiveness and side-effect prevalence–appear to have more muted effects among the most vaccine hesitant respondents. The *prevalence of side-effects*, *effectiveness*, *country of origin* and *vaccine technology* (e.g., mRNA vaccines) determine vaccine acceptance, but they matter little among the vaccine hesitant. Vaccine hesitant people do not find a vaccine more attractive even if it has the most favorable attributes. While the communication of vaccine attributes is important, it is unlikely to convince those who are most vaccine hesitant to get vaccinated.

## Introduction

The development of safe and effective vaccines against the SAR-CoV-2 virus that causes COVID-19 is a "game-changer" in the global fight against the pandemic, especially if rates reach levels sufficient to attain herd immunity [1]. Globally, there are now at least five different vaccines that are widely being used outside of clinical trials: Oxford/Astra-Zeneca, Pfizer-BioNTech, Johnson & Johnson, Moderna, Sinovac, and Sputnik V. To maintain global progress against the COVID-19 pandemic, overcoming COVID-19 vaccine rejection is paramount

**Data Availability Statement:** We pre-registered hypotheses, primary, and secondary analyses before data collection on 26 March 2021 at OSF. We provide our material, data and code on the OSF

project repository (pre-registration link: https://osf.io/ncfbr; project link: https://osf.io/esmdt/).

**Funding:** This project (J.R.) received funding from the European Research Council (ERC) under the European Union's Horizon 2020 research and innovation programme (grant agreement No. 682758). The funders had no role in study design, data collection and analysis, decision to publish, or preparation of the manuscript.

**Competing interests:** The authors have declared that no competing interests exist.

[2]. Despite vast efforts from governments and NGOs, vaccination rates have stalled in some areas where they are widely available [3]. In order to effectively address vaccine rejection, public health experts and policy-makers must understand the determinants of vaccine preferences [4, 5]. Here, we take up a decisive dimension of this challenge: which vaccine attributes do ordinary citizens find attractive, and which attributes are ultimately decisive in determining vaccine acceptance? We also examine differences in the likelihood of uptake across different vaccine attributes for different demographic and attitudinal subgroups, namely people with different levels of vaccine hesitancy and risk preferences.

There is a wide array of vaccine attributes that might affect people's willingness to receive different vaccines. Previous work identifies side effects, effectiveness, and the country of origin [6–15] as factors that are likely to matter. We replicate and expand upon this work with attention to other vaccine attributes such as vaccine technology used (e.g., viral vector, mRNA vaccine), number of already-injected people, time required for large-scale vaccination, and the monetary costs that the vaccine imposes on the society. In particular, attitudes toward new mRNA vaccines remain largely unknown. Motta (2021) finds that the US public do not perceive mRNA vaccines favorably, even though scientists point out that mRNA vaccines are particularly effective and safe. Owen et al. (2021) support this finding, showing that the vaccine type (mRNA vs. weakened virus) is not critical for Canadians' vaccine decisions. However, such a pattern may come from insufficient information about vaccine types in addition to (or instead of) an aversion to mRNA vaccines. Our study addresses this issue by presenting participants with information about specific differences in COVID-19 vaccine technologies.

Our study addresses two additional gaps in the literature. The first one refers to the generalizability of existing research. Specifically, we add data from additional countries (France, Germany, Sweden) during a crucial period when vaccine campaigns had begun but before large portions of the public had been vaccinated. In fact, until shortly before we fielded these surveys, the decision of whether to accept a COVID-19 vaccine was an abstract one. Our conjoint design fielded at the beginning of European vaccination efforts has the benefit of being better informed by the real-world situation at that point in time (which in turn helps to establish the generalizability of earlier conjoint studies, or potentially to demonstrate important contextual differences). Note that we are not alone in this 'generalizability contribution': other research has been published between conducting and publishing our experiment, replicating the effects of side effects, effectiveness, and the country of origin for other countries such as Japan, Canada and the Philippines [14–16].

The second research gap we address is the extent to which individual differences shape the influence of vaccine attributes on COVID-19 vaccine acceptance. Other work already establishes that people with higher vaccine hesitancy and lower risk tolerance may need to be targeted differently by vaccination campaigns than those with higher vaccine favorability and risk tolerance [17]. Furthermore, vaccine hesitancy and risk aversion are associated with real world vaccine behavior [18–21]. Therefore, we evaluate how individual differences in vaccine hesitancy and risk preferences condition the effect of vaccine attributes on vaccine acceptance. In line with this, initial evidence suggests that vaccine hesitancy indeed determines to what extent people are responsive to vaccine attributes. Specifically, a Canadian study has found that people who are willing (vs. not willing) to get a COVID-19 vaccine are more responsive to vaccine attributes such as the country of origin and the effectiveness [15].

Our study sheds light on these factors by conducting an online conjoint experiment on nationally representative quota samples of French, German, and Swedish citizens. Respondents were given pairs of hypothetical vaccine "profiles" that varied across seven dimensions: prevalence of severe side-effects, effectiveness rate, country of origin, vaccine technology, how many people worldwide had already received the (hypothetical) vaccine, how long it would

take to vaccinate the public using that vaccine, and how much the vaccine would cost to the government. Respondents were shown eight trials. For each profile pair, respondents selected vaccine they would choose (41,755 self-reports) and indicated how willing they would be to take each vaccine on an ordinal scale with endpoints labeled as "not at all likely" and "extremely likely" (84,816 self-reports). To examine subgroup variation in reactions to different attributes of vaccines, we split the sample by measures of vaccine hesitancy and risk preferences.

## The effect of vaccine attributes on vaccine acceptance

Several studies examine individual-level predictors of (COVID-19) vaccine acceptance. Women, conservatives, lower-income people, and the highly religious, for example, tend to reject vaccines more [22, 23]. Less work has been dedicated to examining the effects of vaccine characteristics themselves. Much of this research has been undertaken in the context of COVID-19 as competing vaccines have been developed simultaneously, and such has mostly focused on the US (with the exception of work in Brazil, see [24]). People tend to prefer Western-origin vaccines over those developed in Russia or China, and often prefer their own country's vaccine when available (that is, US citizens have preferred US-developed vaccines in these experiments) [12, 25]. Not surprisingly, people also prefer vaccines with greater effectiveness and lower risk of side effects [12, 25, 26]. People also prefer vaccines with full FDA approval over those with full FDA emergency use authorization [10]. Motta (2021) found that US citizens were indifferent to whether the vaccine required one or two doses, time taken for development, or mRNA vs. live virus vaccines. Regarding the vaccine technology, Owen et al. (2021) found a similar pattern, namely that people are indifferent between mRNA or a weakened virus vaccines. Participants in the Motta (2021) and Owen et al. (2021) study were not given information about vaccine types prior to making a choice (whereas our study does provide this information to respondents).

We identify at least three further vaccine attributes that might affect vaccine preferences. When evaluating a specific vaccine, people may prefer a vaccine that has already been administered to a large (vs. small) number of people as this can be interpreted as heuristic about safety [11]. People might also prefer a vaccine that allows faster large-scale vaccination. People might be sensitive to how expensive vaccines are to the government, particularly as COVID-19 has challenged government finances. In fact, assessing the impact of the cost of vaccines is important given that the EU emphasized per-dose price when negotiating contracts with manufacturers (e.g., Guardian 2021; Reuters 2020). We note, though, in our three countries, vaccines are offered at no charge to the recipient. Given the research gap on the effect of these vaccine attributes, we aim to explore to what extent the number of people who received a vaccine, the duration for large-scale coverage, and the costs a vaccine imposes on the society affect vaccine acceptance.

## Vaccine hesitancy, risk preferences, and vaccine acceptance

A rich literature has examined individual-level characteristics correlated with immunization uptake. For instance, vaccine acceptance is relatively high among people who score low in conspiratorial thinking, high in cognitive reflection, and who trust the state, scientists and the healthcare system [23, 27, 28]. In this study, we focus on general vaccine hesitancy and risk preferences.

Vaccine hesitancy, defined as a delay in acceptance or refusal of vaccines despite their availability [29–32], is a major barrier of vaccine acceptance. Vaccine hesitancy is not randomly distributed in the population, a phenomenon that is highly problematic from an

epidemiological perspective [33]. Urged by the need to tackle vaccine hesitancy, research has identified various driving factors, including contextual factors (e.g., media, politics), individual and group factors (e.g., knowledge, social norms), and vaccine-specific factors (e.g., mode of administration, costs) [30, 32, 34].

We expect that people scoring low compared to high on vaccine hesitancy are more favorable towards vaccines in general. However, we also expect vaccine hesitancy to moderate responsiveness to vaccine attributes in terms of overall vaccine preferences, though we lack strong priors on whether higher vaccine hesitancy enhances or attenuates the influence of vaccine attributes on vaccine acceptance. The stronger people with varying levels of vaccine hesitancy differ in their favorability towards (attributes of) vaccines, the higher seems the need for targeted messaging [17]. The role of individual differences in shaping the effect of vaccine attributes on vaccine acceptance is relatively unexplored (see [24, 25] for exceptions), and we hope our work invites further subgroup analyses.

Another body of work looks at the relationship between risk preferences and vaccination. At least in some cases, vaccine hesitancy and general risk aversion are distinct and unassociated with one another [35]. General risk preferences capture tendency to engage in behaviors or activities that are rewarding yet involve some potential for loss [36]. High risk tolerance predicts a series of risky health behaviors, ranging from smoking to avoiding dental care to non-use of seat belts [37]. Recent results suggest that lower risk tolerance is linked to greater vaccine uptake in the case of influenza, as these people may recognize the greater risk of non-vaccination in the face of highly effective vaccines [38]. A complementary perspective argues that vaccination decisions are often made by weighing the probability of getting a disease as well as its severity against the risks that come with inoculation. In particular, people scoring high on general vaccine hesitancy might be more likely to endorse getting a vaccine when the benefit of it is clear and salient, but likely reject vaccines if its benefit is unclear and not salient [39].

So far, it is relatively unexplored whether and how individual differences shape the influence of vaccine attributes on vaccine acceptance. Studies examining conditional effects are essentially a subset of the vaccine attribute studies described above. Gramacho et al. (2021) find that rejection of a Chinese-developed vaccine is particularly strong among those who support President Jair Bolsonaro, for example. Others [25] also show that vaccine efficacy was more influential on acceptance for some subgroups than others in the US (whites, Democrats). Finally, a Canadian study provides initial evidence highlighting that people who intend to get a COVID-19 vaccine are more responsive to vaccine attributes (e.g., country of origin, effectiveness) than people who intend not to get a COVID-19 vaccine [15]. Given that people varying in terms of individual differences such as vaccine hesitancy and risk preferences may need to be targeted differently by vaccination campaigns [17], we highlight the importance of further subgroup analyses.

Overall, we extend research on vaccine acceptance by surveying citizens of France, Germany, and Sweden. With our pre-registered research (link: https://osf.io/esmdt/), we aim to replicate prior work on the effect of vaccine attributes. Specifically, we expect to replicate the strong positive effect of high vaccine effectiveness, few side effects, and a preference for vaccines of German, American and British origin over Chinese and Russian origin. We also expect respondents to prefer less expensive vaccines, vaccines with a larger number of already-injected people, vaccines that take less time to administer at a large scale, and vaccines that are lower-cost. We lack strong priors on responsiveness to vaccine technology used. We also conduct exploratory analyses addressing the question whether vaccine hesitancy and risk preference influence (the effect of vaccine attributes on) vaccine acceptance.

## Methods

### Sample

We conducted online experiments with 5,432 respondents recruited from Dynata in France (*N* = 1,802; April 7–22, 2021), Germany (*N* = 1,799 March 29-April 24, 2021), and Sweden (*N* = 1,831; April 8–23, 2021). Sampling implemented nationally representative quotas for gender, age, and region. Supplementary Table SI1-3 in S1 File provide an overview of the demographics of the three samples. Respondents provided informed consent to participate by clicking "I agree to participate" in the online survey, and their data were anonymized. Respondents were paid a local fee for participating by Dynata. Note that we excluded people that did not fill in any conjoint question.

Note that we considered France, Germany and Sweden as suitable European countries as they had comparable timelines for their vaccine rollouts. In fact, all countries received their first vaccine delivery on the same day (December 27, 2020) as well as administered first vaccinations on the same day (an overview of the development of vaccine uptake of the three countries (in comparison with the EU average vaccine uptake) can be found on the European Center for Prevention and Control's website (https://vaccinetracker.ecdc.europa.eu/public/extensions/COVID-19/vaccine-tracker.html#uptake-tab). Importantly, it is not this research's major aim to examine country-specific differences. Rather, our aim is to gain initial insight into the generalizability across different countries.

Prior to fielding the survey, we conducted a power analysis using the DeclareDesign R package (Version 0.26.0) [40] to determine optimal sample size per country given an α of .05 and 80% power. We used Motta (2021) to inform expected effect sizes for attribute levels. That is, we used an effect size of .05 for 75%, and .12 for 95% effectiveness, .21 for China (country), .06 for UK (country), and .18 for Russia (country). Given our lack of strong priors on the effect sizes of other attribute levels, we assumed the need to detect a small effect (.03). The power analysis with 1,000 iterations revealed that a conjoint experiment with eight trials requires a minimum sample of *N* = 1,500 per country.

### Procedure and conjoint experimental design

Prior to the experiment, respondents filled out demographic information and a battery on baseline vaccine attitudes. Then, respondents were given information about four vaccine types: mRNA vaccines, subunit vaccines, live virus vaccines, and viral vector vaccines. While a previous conjoint experiment [12] did not find an effect of vaccine type, it is unclear whether this is because people are truly indifferent, or whether they lack sufficient information to make a judgement. We gave respondents information to establish that null effects can be attributed to indifference. The exact wording of the provided set of information can be found in the Supplementary Material (see Fig SI8 in S1 File).

Next, respondents were taken to a conjoint choice task. In each of eight scenarios, respondents were presented with two different vaccines. These vaccines differ along seven attributes: (1) side effects, (2) effectiveness, (3) country of origin, (4) vaccine type, (5) vaccinated people, (6) vaccination coverage, and (7) costs (see Table 1 for more information). In each scenario, respondents indicated on a 0–10 scale the raw likelihood of taking such a vaccine (likelihood of uptake) and indicated which of two vaccines they preferred (vaccine choice). Fig SI8 in S1 File depicts a screenshot from the online experiment showing a tabular description of hypothetical vaccines A and B, and the likelihood of uptake and vaccine choice questions. This resulted in 16 self-reported likelihoods of uptake and 8 vaccine choices per respondent, totaling 84,816 self-reported likelihoods of uptake and 41,755 vaccine choices after listwise deletion

**Table 1. Conjoint experimental design.**

| Attribute | Level 1 | Level 2 | Level 3 | Level 4 | Level 5 |
|---|---|---|---|---|---|
| Side effects | 1 in 10,000 | 1 in 100,000 | 1 in 1,000,000 | | |
| Effectiveness | 55% | 75% | 95% | | |
| Country of origin | China | Russia | UK | USA | Germany |
| Vaccine type | Live virus vaccine | Viral vector vaccine | Subunit vaccine | mRNA vaccine | |
| Vaccinated people | 1 million | 10 millions | 100 millions | | |
| Vaccination coverage | In 3 months | In 6 months | In 9 months | | |
| Costs | 10x population EUR/SEK | 50x population EUR/SEK | 100x population EUR/SEK | | |

of missing responses. Missingness was negligible (2.41% for likelihood of uptake, 3.91% for vaccine choice).

## Measures

**Vaccine attributes.**   While every vaccine in the decision scenarios was characterized by the same seven attributes, we randomly altered the levels of the seven attributes. Table 1 provides an overview of the attributes and attribute levels used in our experiments. While two country of origin and vaccine type had five and four attribute levels, respectively, the remaining five attributes had three levels. Given this attribute-level configuration, we had a total of 4,860 different possible vaccine alternatives (5 x 4 x 3^5). Supplementary Tables SI24-26 in S1 File provide an overview of how many times respondents saw vaccines with each attribute level in each sample. We also provide an overview of all questions in the Supplementary Material.

We chose attribute levels to reflect the range of existing COVID-19 vaccines. For instance, the options for *country of origin* attribute matched the origins of approved vaccines such as Sinovac (China), Sputnik V (Russia), Astra-Zeneca (UK), Moderna (USA), and Biontech (Germany). For the *costs* attribute, we assumed a price range between 10 and 100 EUR/SEK (currency unit of the respective country), and multiplied it by the population (as an approximation for the costs related to the vaccination of one's country). The price range was designed to be reasonable given our samples' local currency available price information of COVID-19 vaccines [41].

**Likelihoods of uptake and vaccine choice.**   The main response variables were *self-reported likelihoods of uptake* and *vaccine choice*. Likelihoods of uptake captured how likely respondents think they would choose to receive vaccine A or vaccine B. Specifically, respondents were asked "How likely would you be to choose to receive each of the vaccines?" and could indicate for both vaccines how likely they were to choose them on a scale from 0 ('*not at all likely*') to 10 ('*extremely likely*'). Vaccine choice captured which vaccine respondents chose in hypothetical and randomly assigned A-B choice scenarios. Specifically, respondent could see a tabular description of vaccines A and B and were asked "Which vaccine do you prefer to receive?". Note that we did not force respondents to evaluate the likelihoods of uptake and choice of vaccines. Fig 1 is a screenshot of the vaccine decision scenario and outcomes questions that respondents received.

**Vaccine hesitancy.**   To test whether vaccine acceptance varies across people that differ in terms of vaccine hesitancy, we asked respondents to fill in seven items from the parental perspectives regarding vaccines scale [29, 31] plus one novel item. Items were answered on a 5-Point scale from '*strongly disagree*' (1) to '*strongly agree*' (5). The reliability (α) was satisfactory for all three samples (France = .75, Germany = .81, Sweden = .77). Note that we re-coded

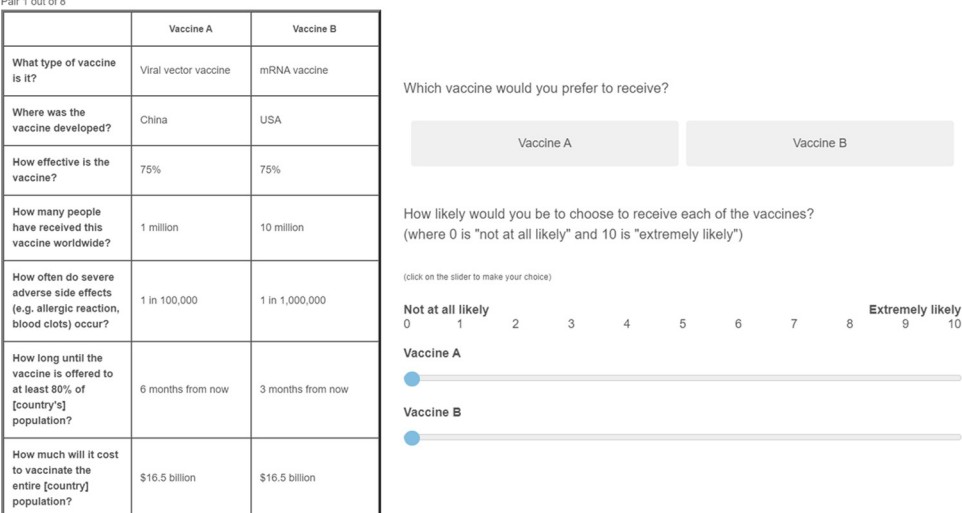

**Fig 1. Screen shot of a vaccine decision scenario (translated into English).** Before being exposed to eight vaccine decision scenarios, respondents were exposed to (1) general information about vaccines, and (2) descriptions of the different vaccine types (see Supporting information).

the items in a way that higher scores indicate higher vaccine hesitancy. We also used latent class analysis (LCA; Mplus Version 8.5) to identify three vaccine attitude types that we use in the subgroup analysis: strong vaccine supporters (n = 2,033 / trials = 32,238), reserved or concerned vaccine acceptors (n = 1,931 / trials = 19,734), and the vaccine hesitant (n = 1,254 / trials = 19,734). Note that fit indices indicate that a three class solution (log-likelihood (LL) = -40,515.63, Akaike Information Criterion (AIC) = 81,187.27, Bayesian Information Criterion (BIC) = 81,702.06; Sample-Size Adjusted BIC (ssBIC) = 81,454.20; Lo-Mendell-Rubin adjusted Likelihood Ratio Test (LMR-LRT) = 2,557.73 ($p$ = 0.001)) fits our data best (four class solution: LL = -39,658.16, AIC = 79,526.33, BIC = 80,219.31; ssBIC = 79,885.66; LMR-LRT = 1,707.59 ($p$ = 0.001)).

**Risk preferences.** To test whether vaccine acceptance varies across people that differ in terms of their risk preferences, we asked respondents to indicate on a single item their general tendency to take risks. The general risk preference question was answered on a scale from *'not at all willing to take risks'* (0) to *'very willing to take risks'* (10). Higher scores indicate higher risk tolerance. For the sub-group analysis, we define those risk seeking as greater than the scale mid-point (n = 2,377 / trials = 37,630), and those at the midpoint or below are coded as risk-averse (n = 2,834 / trials = 44,674).

**Ethics.** We obtained ethical approval for this study from the Ethics Committee of the College of Social Sciences and International Studies of the University of Exeter on 23 December 2020 (reference number of ethical approval: 202021–045). This research complies with General Data Protection Regulation requirements. The data were collected, and made available on OSF without identifying information, and with informed consent from the respondents.

**Pre-registration, deviations, and researcher degrees of freedom.** We pre-registered hypotheses, primary, and secondary analyses before data collection on 26 March 2021 at OSF. We provide our material, data and code on the OSF project repository (pre-registration link: https://osf.io/ncfbr; project link: https://osf.io/esmdt/).

First, while our pre-registration clearly states that we will perform analyses for the average marginal component effects (AMCEs) and marginal means (MMs) for the binary choice outcome, we neglected to include language stating that we would perform the same procedure for

the ordinal likelihood of uptake outcomes, which we present MMs for in the main text here for ease of interpretation (see [42–44]). For full transparency, we have the full slate of AMCE and MM analyses for both the binary choice task and ordinal likelihood of uptake items in the Supporting information (Fig SI1-3 in S1 File, Tables SI4-7 in S1 File.) We do note, though, that substantive results are identical across model choice (MMs vs. AMCEs) and outcome variable used (binary vaccine choice vs. likelihood of uptake). Second, our preregistration discussed using data collected from Hungary. These data were not available during analysis, and will instead be presented elsewhere. More importantly, the experimental design of the Hungary experiment deviated from the other experiments in that participants were not presented with information about vaccine technologies. The Hungary conjoint was also embedded in a larger survey dedicated to other topics. Finally, Hungary used a wider array of vaccines (including those from Russia and China). For these reasons, we have decided to report Hungary results elsewhere.

Regarding researcher degrees of freedom, the pre-registration left several choices to be made after data collection. Our preregistration did not explicitly state whether we would pool data from the three countries. Our power analysis is sufficient to look at each country separately (at least for main effects). Given the strong similarity of results across the three countries, we have decided to pool them into a single analysis, but retain country specific results in the Supporting information. By pooling the three countries, our subgroup analyses are better powered. While we pre-registered our intent to perform exploratory analysis of the conjoint for different subgroups (pre-treatment vaccine attitudes, risk preferences, numeracy, and others), exactly how we would identify or code these groups was left until after data collection. For vaccine hesitancy, we have opted for a latent class model with three instead of five (as preregistered) classes. For risk tolerance, we have opted for a simple dichotomous split with those above the scale midpoint coded as risk seeking, and those at or below the scale midpoint coded as risk averse. Focusing on three and two (vs. more) subgroups has the advantage of a simple and clear interpretation. Furthermore, the hypothesis concerning vaccine hesitancy does not focus on the multidimensionality of vaccine attitudes, but rather centers on how people of placed on the lower and upper end of the vaccine hesitancy scale differ in terms of responsiveness to vaccine attributes. For this purpose, considering three (vs. five) vaccine hesitancy subgroups is more straightforward.

Consistent with the preregistration, we did additional exploratory analysis with other subgroups, particularly related to cognitive reflection and numeracy. For simplicity, we have kept this exploratory analysis out of the current manuscript. In summary, we found that those higher in cognitive reflection or numeric ability appear slightly more responsive to attributes that rely on numeric information.

## Results

### Identifying favorable vaccine attribute levels

We computed marginal means (MMs) for vaccine attribute levels. MMs describe the average self-reported likelihood of a vaccine being accepted, on a 0–10 scale, when that vaccine has an attribute at a particular level (e.g., 95% effectiveness). We report MMs as they illustrate the baseline level of vaccine uptake, and allow for the comparison of vaccine uptake between the full sample and different subgroups. We preregistered our analyses as average marginal component effects (AMCEs), which estimate treatment effects when vaccine attributes are altered relative to a reference level (e.g., from 55% effectiveness to 95% effectiveness), and are available in the Supplementary Material (Fig SI1 in S1 File), as are full numerical estimates of both MMs and AMCEs (Tables SI4-5 in S1 File) [42–44]. We present the MMs in the main text to

facilitate interpretation that is not predicated on a specific reference category [44]; similarly, we choose to present the vaccination likelihood as the primary outcome (versus the binary choice measure) to more clearly demonstrate levels of support for different vaccine profiles. Substantive results are identical across model choice (MMs vs. AMCEs) and outcome variable (vaccine uptake likelihood vs. binary vaccine choice).

Fig 2 plots MMs from the full sample of self-reports of how likely they would be to receive a vaccine with particular attribute levels across all respondents in all three countries, on the original 0–10 scale. Note that the dashed line represents the grand mean (i.e. overall mean across all attributes) and is useful for interpretation: Values above the line indicate attribute levels that increase profile favorability and values below indicate attribute levels that decrease favorability (compared to the average of all profiles). Here, we focus our attention on vaccine attributes with the strongest effects on respondents' likelihood of accepting a vaccine: country of origin of a vaccine, effectiveness, and side effect frequency. Our conjoint experiments also ask respondents to make a binary choice between the vaccines they are shown. Results for this outcome variable are substantively similar to the results for our main outcome (Fig SI2-3 in S1 File, Tables SI6-7 in S1 File).

The MMs for country of origin show that uptake was more likely than not when vaccines originated from the UK (MM = 5.33, 95% CI = [5.25–5.40]), the USA (5.35, [5.28–5.43]), or especially Germany (5.52, [5.44–5.60]), whereas this was not the case for Russian (4.82, [4.74–4.90]) or Chinese vaccines (4.64, [4.56–4.72]). Similarly, respondents were less likely to accept a vaccine with an effectiveness rate of 55% (4.68, [4.61–4.75]), but more likely to accept one with 95% effectiveness (5.57, [5.50–5.64]). The likelihood of uptake for a vaccine with an effectiveness rate of 75% (5.15, [5.08–5.22]) was average. In both cases, going from the least favorable (China/55% effectiveness) to the most favorable attribute level (Germany/95% effectiveness) brought about a similar change in the average likelihood of accepting the vaccine, of just under 0.9 points (a 19% increase). Side effect prevalence shows a similar pattern of favorability, but this is less pronounced, with rates of '1 in 10,000' slightly unfavorable (4.94, [4.87–5.01]), '1 in 100,000' (5.16, [5.09–5.23]) average, and '1 in 1,000,000' (5.30, [5.22–5.37]) more favorable. Going from the least favorable ('1 in 10,000') to the most favorable attribute level ('1,000,000') brought a 0.4 point change in the average likelihood of accepting the vaccine (a 8% increase). Again, note that AMCEs (i.e. effect sizes) estimate treatment effects when the MMs for vaccine attributes are altered relative to a reference level (see Fig SI1 in S1 File).

Compared to these strong effects of country of origin, side effect frequency and effectiveness, the remaining vaccine attributes affect the likelihood of uptake very little. Respondents were slightly more favorable towards vaccines that provide coverage of the population more quickly, that cost less, and that have already been used on more people. Notably, far from being skeptical of the newly innovated mRNA vaccines, respondents actually slightly preferred these relative to more established types.

Our findings are robust to restricting the sample to respondents who passed attention checks (see Fig SI4 in S1 File and Table SI8-11 in S1 File). There were minimal cross-country differences in the effects of vaccine attributes, with the exception of German-developed vaccines being particularly well received in Germany. MMs revealed, however, that French (compared to German and Swedish) respondents reported relatively low overall favorability towards vaccines. That is, French respondents' reported likelihood of uptake was estimated to be 4.70 (4.57–4.82) for vaccines with '1 in 10,000 side effects', 4.92 (4.80–5.05) for vaccines with '1 in 1,000,000 side effects', 4.40 (4.27–4.52) for vaccines with '55% effectiveness', 5.21 (5.08–5.34) for vaccines with '95% effectiveness', 5.13 (4.99–5.27) for vaccines from Germany, and 4.31 (4.17–4.45) for vaccines from China. In contrast, German / Swedish respondents' reported likelihood of uptake was estimated to be 5.00 (4.88–5.12) / 5.12 (5.00–5.24) for

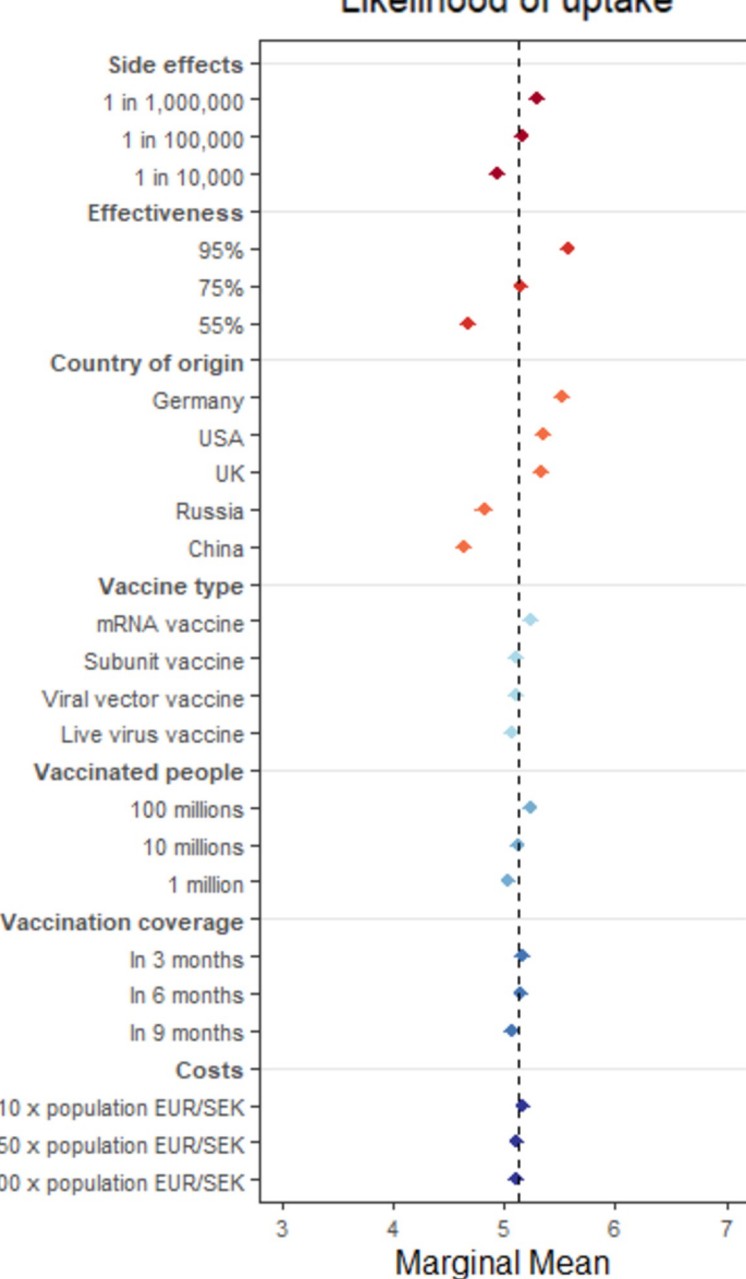

**Fig 2. MMs for self-reported likelihood of uptake.** The figure reports the marginal mean point estimates are plotted with 95% CIs, representing the average likelihood of uptake at each vaccine attribute level. The dashed line represents the grand mean (5.11).

vaccines with '1 in 10,000 side effects', 5.42 (5.30–5.54) / 5.54 (5.42–5.66) for vaccines with '1 in 1,000,000 side effects', 4.71 (4.59–4.83) / 4.93 (4.81–5.05) for vaccines with '55% effectiveness', 5.73 (5.61–5.85) / 5.78 (5.66–5.90) for vaccines with '95% effectiveness', 5.79 (5.66–5.92) / 5.64 (5.51–5.77) for vaccines from Germany, and, 4.71 (4.57–4.85) / 4.91 (4.77–5.04) for vaccines from China (see Fig SI5 in S1 File and Table SI12-15 in S1 File).

## Subgroup analyses: Vaccine hesitancy, and risk preference

To examine the moderating role of vaccine hesitancy and risk preferences on the effects of vaccine attribute on vaccine preferences, we estimated AMCEs and MMs for different subgroups on these dimensions. Just as with the main results, we depict MMs here, but subgroup AMCEs can be found in the Supporting information. We do note, though, that AMCEs, while useful for estimating the causal effect of attributes within subgroups, are subject to inferential errors when considering relative preferences across groups, a problem not encountered with MMs [44]. Unless otherwise indicated, we only describe differences of MMs where we also found a corresponding statistically significant AMCE of the attribute within subgroups (see Table SI17 in S1 File).

For vaccine hesitancy, we separated the sample through conducting a LCA on the 7-item battery of vaccine attitudes to identify groups of people with similar profiles of vaccine attitudes. Our LCA produced a three-group solution: strong vaccine supporters (38%), reserved (or concerned) vaccine acceptors (37%), and the vaccine hesitant (25%). MMs for each of these groups are plotted in Fig 3. In absolute terms, vaccine supporters are consistently more favorable to vaccines than reserved vaccine acceptors and the vaccine hesitant. In fact, vaccine supporters were substantially more likely to accept the vaccines with objectively the worst attributes–'1 in 10,000' side effects (5.64, [5.53–5.74]) or 55% effectiveness (5.30, [5.19–5.40])–than the vaccine hesitant were to accept the vaccines with objectively the best attributes–'1 in 1,000,000' side effects (4.39, [4.23–4.56]) or 95% effectiveness (4.62, [4.46–4.79]). This pattern is also observed across the other vaccine attributes: type, number already vaccinated, speed of population coverage, and cost. (Note that full numerical estimates of MMs and AMCEs (effect size) for likelihood of uptake and vaccine choice as well as a subgroup MM plot for vaccine choice are provided in the Supplementary Material (Fig SI6 in S1 File, Tables SI16-19 in S1 File).)

Additionally, we find that the preferences of subgroups higher in vaccine hesitancy are less responsive to vaccine attributes. For example, vaccine supporters are much more favorable towards vaccines of German origin (6.56, [6.45–6.66]) than they are towards those of Chinese origin (5.22, [5.10–5.34]) (a 26% increase), but this difference is smaller for vaccine acceptors (a 19% increase), and it is comparatively negligible for the vaccine hesitant (a 7% increase). Similarly, vaccine supporters are substantially more likely to take a vaccine with 95% (6.51, [6.41–6.62]) as opposed to 75% effectiveness (5.97, [5.86–6.10]), which is in turn more favorable than 55% effectiveness (5.29, [5.19–5.40]). This patterns holds for vaccine acceptors, though to a slightly lesser extent. For the vaccine hesitant, these differences are again much smaller, as is evident from comparing the variance within subgroups in Fig 3. Going from the least favorable (55% effectiveness) to the most favorable attribute level (95% effectiveness) brought a 1.2 point change in the average likelihood of accepting the vaccine for vaccine supporters (a 23% increase), a 0.8 point change for vaccine acceptors (a 18% increase), and a 0.5 point change in for vaccine hesitants (a 11% increase). Finally, vaccine supporters clearly favor vaccines with rarer side effects, and there is a smaller effect in the same direction for vaccine acceptors, but the vaccine hesitant show an inconsistent pattern of favorability regarding side effects (see Table SI16-17 in S1 File). Such differences in effects are far less apparent across the remaining attributes because the effects of these attributes are small to begin with.

We also examined whether the effects of the vaccine attributes vary across people with different levels of risk preferences. To capture risk preferences, respondents reported on a scale from 0 to 10 to what extent they avoid (0), or seek risks (10). We used the scale midpoint to split people into a risk-averse and risk-seeking group. In absolute terms, risk-seeking respondents are consistently more favorable to vaccines than risk-averse respondents. To a large

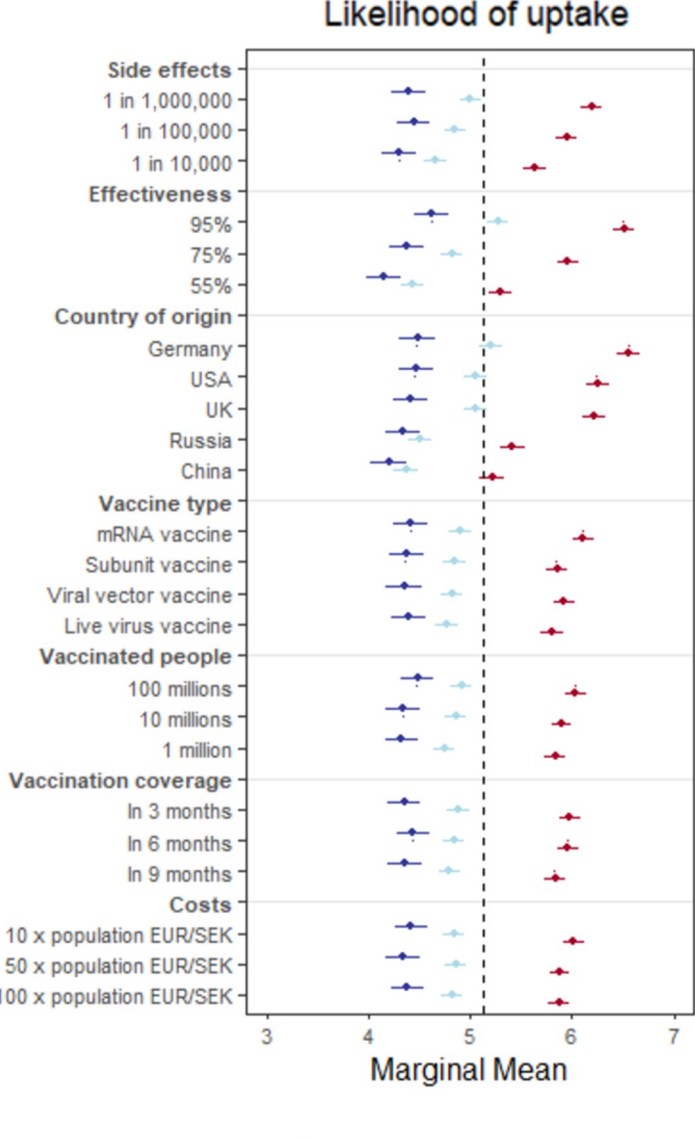

**Fig 3. Subgroup analysis: Differences across different vaccine hesitant groups.** Each dot and error bar represents the MM (and its 95% CI) of vaccine attributes on self-reported likelihood of uptake for the three groups of people with varying vaccine attitudes. The dashed line represents the grand mean.

extent, risk-seeking respondents are as likely to accept even the least optimal vaccines than risk-averse respondence are to accept the most optimal vaccines (see Fig 4). For example, risk seekers are more likely to accept a vaccine with '1 in 10,000' chance of severe side effects (5.49, [5.39–5.60]) than the risk-averse are to accept a vaccine with an order magnitude decrease ('1 in 1,000,000') chance for a severe side effect (4.91, [4.81–5.01]). Risk seekers are also approximately as likely to accept a vaccine with 55% effectiveness (5.19, [5.09–5.30]) as the risk-averse are to accept a vaccine with 95% effectiveness (5.17, [5.07–5.27]), and approximately as likely to accept a Chinese vaccine (5.16, [5.04–5.28]) as the risk-averse are to accept a German vaccine (5.15, [5.04–5.26]). (Full numerical estimates of MMs and AMCEs (effect sizes) for

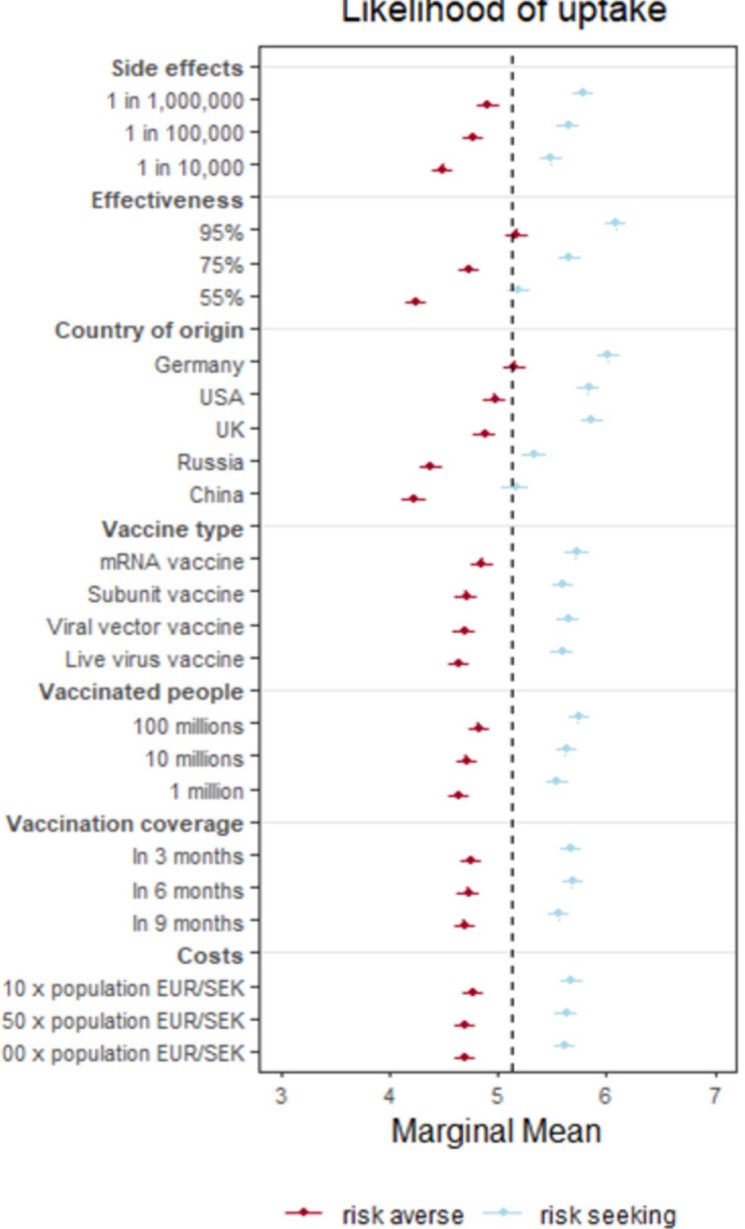

**Fig 4. Subgroup analysis: Differences across different risk preference groups.** Each dot and error bar represents the MM (and its 95% CI) of vaccine attributes on self-reported likelihood of uptake for the two groups of people with varying risk preferences. The dashed line represents the grand mean.

likelihood of uptake and vaccine choice as well as a subgroup MM plot for vaccine choice are provided in the Supplementary Material (Fig SI7 in S1 File, Tables SI20-23 in S1 File).)

Unlike vaccine hesitancy, however, risk perceptions do not meaningfully moderate the effect of vaccine attributes on vaccine acceptance. For example, people scoring low and high on risk preferences are similarly more favorable towards vaccines of German origin than they are towards those of Chinese origin. The difference in MMs is approximately 0.9 points for both risk-averse and risk-seeking people (0.93 and 0.86) (Again, we only describe differences of MMs where we also found statistically significant AMCEs for these differences (see

Table SI21 in S1 File)). Similarly, risk-averse and risk-seeking people are roughly equally more likely to take a vaccine with 95% as opposed to 75% (risk-averse difference approx. 0.44, risk-seeking 0.43), and 55% effectiveness (risk-averse difference approx. 0.92, risk-seeking 0.89). Finally, risk-averse and risk-seeking people similarly favor vaccines with rarer side effects. Similar tendencies hold, both absolutely and relatively, for the other vaccine attributes. These findings indicate that risk acceptance affects baseline favorability towards vaccines, but not responsiveness to vaccine attributes.

## Discussion

To better understand why people prefer one particular COVID-19 vaccine over another, and which vaccine attributes determine vaccine acceptance, we conducted online conjoint experiments with broadly representative samples of citizens from France, Germany, and Sweden choosing between eight pairs of hypothetical COVID-19 vaccines that randomly varied across seven attributes. Replicating previous work [11–16, 25, 26, 45], we found that people strongly prefer vaccines with few side effects, with high effectiveness, and that were developed in western countries. People also prefer vaccines that have already been injected to many people, require only a short time for large-scale vaccination coverage, and are low in cost, but these attributes mean less to the public than side effects, effectiveness, or national origin. We interpret our finding that people prefer vaccines that many others have previously taken as motivation to mitigate risk in vaccine choice [5, 9, 46]. If more people have been given a specific vaccine with a high level of effectiveness and few side effects, the vaccine is more clearly safe than if only a handful have taken the vaccine. The effects of vaccine attributes are similar across France, Germany, and Sweden as well as other countries from existing research (e.g., in the US, Canada, Latin America, or Japan; see [12–15]). Our findings also show clear - albeit small - preferences for mRNA vaccines. (Due to statistical power considerations, we did not randomize access to this information. Consequently, we are unable to discern whether the information we provide or the more general information environment surrounding around mRNA vaccines–or both–may contribute to our result.) Motta (2021) and Owen et al. (2021) did not find an effect for vaccine type, but also did not give respondents information on different vaccine types prior to the conjoint.

Finally, we demonstrate that there are significant individual-level differences in terms of vaccine acceptance. The higher people score on general vaccine hesitancy and the lower they score on risk tolerance the less favorable they are towards vaccines. While this pattern is in line with existing evidence [15, 30, 32, 34, 39], our data also show that the higher people score on general vaccine hesitancy the less they distinguish between vaccine attributes when choosing vaccines. Though this study does not test underlying psychological mechanisms of this pattern, these findings suggest that for people to weigh distinctions between vaccines, they must believe in vaccination as a worthy preventative measure first. Although speculative, it seems likely that only people that accept and/or support vaccines are willing to allocate cognitive resources to process information about vaccine attributes (e.g., side effects, country of origin). The vaccine hesitant are less likely to be willing to cognitively engage with information about vaccine options as they are negative towards vaccines anyway, independent of any vaccine attributes. Another potential mechanism is that people who are generally positive (vs. negative) towards vaccines are more experienced and hence better in distinguishing between different vaccine options characterized by vaccine attributes (e.g., different levels of effectiveness and side effects). Similarly, it might be that more risk averse people tend to allocate more cognitive resources to and are more experienced in processing information about different vaccine attributes. Regardless of the mechanism, our findings emphasize the importance of considering

individual-level differences when predicting and explaining why some people prefer one COVID-19 vaccine over another.

This research has four primary limitations. First, we did not provide respondents with definitions of our vaccine attributes, and we did also not measure the meaning of the tested vaccine attributes. Hence, we do not know to what extent a common understanding of vaccine attributes such as side effects and effectiveness exists. Future research should be more aware of this aspect (see [47]).

Second, respondents made hypothetical choices about hypothetical vaccines. We recorded self-reported likelihood of uptake and vaccine choice. While there is evidence emphasizing that vaccine intentions are strongly related to actual vaccine behavior [12, 48–50], we need to keep in mind that self-reports are generally known to explain only a moderate amount of variance in actual behavior [51], and that people may make different decisions (especially in not choosing any vaccine) when researchers are not there to observe vaccine decisions [43, 52]. Even though we have to assume that our data is biased by respondents' tendency to report in a socially desirable manner, our experimental design decreases the problem of social desirability bias as this bias is the same across all our conjoint scenarios.

Third, the attributes respondents chose between did not fully mirror those of real-world vaccines. For example, we defined the levels for the costs as EUR 10 per person (i.e., €10 × the country population, or the SEK equivalent), EUR 50 per person, or EUR 100 person. The actual costs for the vaccine are typically less than EUR 20 person in the EU [53], though globally prices may vary. Vaccine cost has not been a significant aspect of discourse around vaccination efforts in our sample countries. While inclusion of elements outside standard discourse (and with levels that do not reflect real world options) puts some limits on generalizability, we believe that the description of our hypothetical vaccines reflect real-world options relatively well.

Fourth, neither the Russian Sputnik V nor the Chinese Sinovac vaccine are (currently) approved for use in France, Germany, and Sweden. We cannot rule out that this affects the way how respondents view the hypothetical vaccines from China and Russia in the conjoint experiment. Yet, it should be noted that Motta (2021) found a similar pattern in an experiment conducted in the US before there were *any* approved COVID-19 vaccines. It is also possible that the Russian and Chinese vaccines have not been approved because policymakers share some of the same beliefs as the public, or at least anticipate the public opposition that we document, and hence do not want to take the risk of approving them.

Regarding the lack of ecological validity of our study, it is noteworthy that our findings are in line with more recent research focusing on the real-world relevance of the role of different vaccine profiles. In fact, Merkley and Loewen revealed that people have varying preferences for real-world vaccines with different safety and efficacy profiles. For instance, people prefer the Pfizer-BioNTech and Moderna vaccine compared to the Astra-Zeneca and Johnson & Johnson vaccine as they perceive them as more safe and effective [47, 54].

The present findings can inform public health experts and policy makers on how to design effective vaccine campaigns. Respondents clearly prefer vaccines with side effects that occur at a rate lower than '1 in 1,000,000', vaccines with at least 95% effectiveness, and western vaccines. Yet, real-world COVID-19 vaccines do often not meet these benchmarks. Therefore, it is important to boost citizen's ability to realize that the benefits of currently-available vaccines outweigh any costs [55, 56]. Boosting citizen's general risk-illiteracy, for example, could help them perform more rational cost-benefit evaluations. In fact, training citizen's (numerical) ability to interpret numerical information such as side effects, and effectiveness could prevent them from overestimating the risk of safe vaccines [55, 57–59].

Most importantly, our findings highlight the importance of micro-targeting vaccination messages to different sections of the public. For groups that broadly accept vaccines, messaging that makes a detailed accounting of the safety and effectiveness of the vaccine(s) available can be effective in encouraging vaccination. However, for groups that are broadly vaccine-hesitant and risk-intolerant, messaging needs to concentrate more on changing attitudes towards vaccines, and make it clear and salient that the benefit of getting a COVID-19 vaccine outweighs the costs [17, 39, 60].

Another lever that seems important for vaccine campaigns is the communication of the country of origin. Our results imply that people prefer vaccines developed in western countries. Since vaccines of western origin are available in most countries, highlighting the origin of the vaccine (or parts of the vaccine) can encourage adoption. If Germans' strong preference for German vaccines is any indication, emphasizing the role of one's own country in developing an available vaccine can also be profitable. Importantly, this country of origin effect also occurs outside of Europe and the US such as for instance in Brazil [24], meaning such messaging can be profitable in most sections of the world.

Finally, we highlight that even though we emphasize the importance of micro-targeting vaccination messages, we cannot make predictions about the effectiveness of such an approach. Future research needs to systematically test this approach. Given the varying availability of vaccines and the varying vaccine policies and campaigns across countries, it would also be interesting to test how such contextual factors determine the effectiveness of micro-targeting vaccination messages.

## Supporting information

**S1 File.**
(ZIP)

## Acknowledgments

We are grateful to the two anonymous reviewers for their comments and feedback. The authors declare that there is no conflict of interest.

## Author Contributions

**Conceptualization:** Sabrina Stöckli, Anna Katharina Spälti, Joseph Phillips, Florian Stoeckel, Benjamin Lyons, Vittorio Mérola, Paula Szewach, Jason Reifler.

**Data curation:** Sabrina Stöckli, Anna Katharina Spälti, Joseph Phillips, Florian Stoeckel, Matthew Barnfield, Jack Thompson, Benjamin Lyons, Vittorio Mérola, Paula Szewach, Jason Reifler.

**Formal analysis:** Sabrina Stöckli, Anna Katharina Spälti, Joseph Phillips, Florian Stoeckel, Matthew Barnfield, Jack Thompson, Benjamin Lyons, Vittorio Mérola, Paula Szewach, Jason Reifler.

**Funding acquisition:** Jason Reifler.

**Investigation:** Sabrina Stöckli, Anna Katharina Spälti, Joseph Phillips, Florian Stoeckel, Matthew Barnfield, Jack Thompson, Benjamin Lyons, Vittorio Mérola, Paula Szewach, Jason Reifler.

**Methodology:** Sabrina Stöckli, Anna Katharina Spälti, Joseph Phillips, Florian Stoeckel, Matthew Barnfield, Jack Thompson, Benjamin Lyons, Vittorio Mérola, Paula Szewach, Jason Reifler.

**Project administration:** Sabrina Stöckli, Anna Katharina Spälti, Joseph Phillips, Florian Stoeckel, Matthew Barnfield, Jack Thompson, Benjamin Lyons, Vittorio Mérola, Paula Szewach, Jason Reifler.

**Visualization:** Sabrina Stöckli, Joseph Phillips, Florian Stoeckel, Matthew Barnfield, Jack Thompson, Benjamin Lyons, Vittorio Mérola, Paula Szewach, Jason Reifler.

**Writing – original draft:** Sabrina Stöckli, Anna Katharina Spälti, Joseph Phillips, Florian Stoeckel, Matthew Barnfield, Jack Thompson, Benjamin Lyons, Vittorio Mérola, Paula Szewach, Jason Reifler.

**Writing – review & editing:** Sabrina Stöckli, Joseph Phillips, Florian Stoeckel, Matthew Barnfield, Jack Thompson, Benjamin Lyons, Vittorio Mérola, Paula Szewach, Jason Reifler.

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
