## [Decision Letter · Decision Letter 0]

15 Dec 2021

PONE-D-21-35451Which vaccine attributes foster vaccine uptake? A cross-country conjoint experimentPLOS ONE

Dear Dr. Stöckli,

Thank you for submitting your manuscript to PLOS ONE. After careful consideration, we feel that it has merit but does not fully meet PLOS ONE’s publication criteria as it currently stands. Therefore, we invite you to submit a revised version of the manuscript that addresses the points raised during the review process.

Reviewers provide clear comments, questions and suggestions. I invite you to respond to all of them. On top of their evaluations, I would like to invite you to respond to my three points detailed below.  

The decision to focus on three groups for the LCA comes a bit out of the blue without justification (line 242/243, 285/86, 359/60). Reading the pre-registered analysis plan, I see this vague mention to *five* groups: “(Based on other work by *author*, we anticipate using this scale to create a latent class model with five classes: strong vaccine supporter, supporter with reservations, vaccine hesitant, "anti-vax", and measurement error.)” I am puzzled. Can you please clarify/explain this gap (5 v/s 3 groups) and provide more information on, for example, the model fit indicators?

On the deviation from the preregistered analysis plan. I would *not* require you to include Hungary, but of course the paper would benefit from including this sample. Would it be possible to include the sample at this stage? 

There might be a social desirability bias with the self-reported likelihood of vaccine uptake. Although you do not mention ‘social desirability bias’, this is what you probably refer to on lines 459-64. The discussion of this limitation seems, however, quite weak. I have a very partial view as I believe that there is much stronger evidence of a bias than not (Daoust et al. 2020, 2021; but see Larsen et al. 2020 for null results and Muzert and Selb 2020 for mixed findings) and I do not believe that claiming that the “vaccine intentions strongly related to actual vaccine behavior” is the best defence for the self-reported measure. First, I would tend to think that you should acknowledge that there is at least a substantial risk that the means are inflated by a social desirability bias. Second, and this is key, I believe that your research design makes the bias less problematic because the social desirability bias is very likely homogenous across the different treatments — allowing you to compare the differences without systematic bias.

We look forward to receiving your revised manuscript.

Kind regards,

Jean-François Daoust

Academic Editor

PLOS ONE

Journal Requirements:

2. Please provide additional details regarding participant consent. In the Methods section, please ensure that you have specified (1) whether consent was informed and (2) what type you obtained (for instance, written or verbal). If your study included minors, state whether you obtained consent from parents or guardians. If the need for consent was waived by the ethics committee, please include this information.

[This project received funding from the European Research Council (ERC) under the European Union’s Horizon 2020 research and innovation programme (grant agreement No. 682758). The authors declare that there is no conflict of interest.]

 [This project (J.R.) received funding from the European Research Council (ERC) under the European Union’s Horizon 2020 research and innovation programme (grant agreement No. 682758). The funders had no role in study design, data collection and analysis, decision to publish, or preparation of the manuscript.]

4. Please ensure that you refer to Figure 3 in your text as, if accepted, production will need this reference to link the reader to the figure.

Reviewers' comments:

Reviewer's Responses to Questions

**Comments to the Author**

1. Is the manuscript technically sound, and do the data support the conclusions?

Reviewer #1: Yes

Reviewer #2: Yes

2. Has the statistical analysis been performed appropriately and rigorously? 

Reviewer #1: Yes

Reviewer #2: Yes

3. Have the authors made all data underlying the findings in their manuscript fully available?

Reviewer #1: Yes

Reviewer #2: Yes

4. Is the manuscript presented in an intelligible fashion and written in standard English?

Reviewer #1: Yes

Reviewer #2: Yes

5. Review Comments to the Author

Reviewer #1: This paper challenges a critical issue (preference on COVID-19 vaccine) with plausible and recommended empirical practice (pre-registration, open data, and ethical approval).

Moreover, PLOS ONE is relevant to the paper because broad readers should reach and discuss their findings.

I then suggest accepting the paper with a minor revision.

My suggestion is to update the literature review part.

This paper's contribution is international comparison and generalizability with multiple survey sites (France, Germany, Sweden). However, the introduction discussed only findings in a few countries. The survey experiments on COVID-19 vaccine is rapidly increasing and reports finding in various countries (including. for instance, Malawi (Kao, et., al 2021) and Netherlands (Reeskens, et., al. 2021)). I then recommend that the authors update the literature review part, including recent results of survey experiments.

Kao, K., Lust, E., Dulani, B., Ferree, K. E., Harris, A. S., & Metheney, E. (2021). The ABCs of Covid-19 prevention in Malawi: Authority, benefits, and costs of compliance. World development, 137, 105167.

Reeskens, T., Roosma, F., & Wanders, E. (2021). The perceived deservingness of COVID-19 healthcare in the Netherlands: a conjoint experiment on priority access to intensive care and vaccination. BMC public health, 21(1), 1-8.

Reviewer #2: The manuscript “Which vaccine attributes foster vaccine uptake? A cross-country conjoint experiment” provides an empirical test of the characteristics people value in COVID-19 vaccines by using a multi-country conjoint study. The work is similar to what has been conducted previously by Motta (2021) in the U.S. and by the Media Ecosystem Observatory in Canada, but helpfully extends on these analyses in several ways.

A credit to the authors: this is a well-executed project and an exceptionally polished manuscript. It absolutely should be published. I only have very minor points for what they are worth.

• Might be worth looking at a public report conducted by the Media Ecosystem Observatory in Canada. They conduct a conjoint experiment, much like Motta (2021) and the authors, that looks at sub-group effects in vaccine attributes in Canada. They find similar results: people who indicate the will take a COVID-19 vaccine are more responsive to these characteristics than those who report they will not or who are uncertain https://www.mcgill.ca/maxbellschool/files/maxbellschool/meo_vaccine_hesistancy_1.pdf

• A couple other studies provide nice support for the authors’ findings using real-world vaccines. A manuscript by Merkley and Loewen (https://osf.io/ng9qh/) find that people are more likely to differentiate by COVID-19 vaccine brand – as in, reluctance to take AstraZeneca and Johnson & Johnson compared to Pfizer and Moderna – if they are supportive of vaccines, and another article by these authors (JAMA Network Open, 2021) shows people are persuaded to take less preferred vaccines (like AZ and JJ) by highlighting their capacity to prevent death

• Why France, Germany and Sweden? I’d like to see some justification of the case selection. It might also be useful to provide a brief rundown of the vaccine rollout differences across these cases in terms of timeline and vaccine availability. I encourage the authors to at least include the country-by-country MMs in the main text for readers. I think it’s worth highlighting the unique cross-national scope of this study that gets glossed over by only presenting pooled results. If the results are similar, that’s interesting!

• I wonder about the choice of attributes for side effects and effectiveness. Do people have a common understanding of the former? Or are some people thinking “fever” while others are thinking VITT? I recognize other studies have taken a similar approach, but the comparative weakness of the effect for side effects may be due to this lack of precision (and possibly R’s numeracy). As we saw with the VITT fiasco, people really do seem to be responsive to side effect issues

• Similarly with effectiveness there is no clarity for respondents if the meaning is “symptomatic and asymptomatic infection” or just symptomatic infection or hospitalization or death. The Merkley and Loewen (2021) piece suggests people are responsive to symptomatic infection prevention and death prevention characteristics. Respondents may not be entirely clear on what is meant by effectiveness. Obviously, the authors cannot re-run these experiments, but I think it’s worth pointing out these limitations and proposing new research unpacks the side effect and effectiveness characteristics.

• I’d like to see a picture of the conjoint task in the main text

• I’d like to see some more effort to highlight effect sizes. These differences we observe in the means, how sizable are they? Might be useful to express differences in standard deviations and compare them to the sizes that have been found in previous work

• These findings, in addition to the works pointed to above, lead to some interesting questions that the authors only start to hint at in their discussion. What are the mechanisms behind these sub-group differences? And what exactly do we make of the responsiveness of vaccine supporters to these vaccine characteristics? Would we really see slippage for vaccines with effectiveness in the 50 percent range in the real world? Or do we only really observe this because respondents are aware there are other available vaccines with superior characteristics? I think it’s worth thinking about whether these sub-group effects would generalize to other areas with less vaccine choice.

6. PLOS authors have the option to publish the peer review history of their article (what does this mean?). If published, this will include your full peer review and any attached files.

Reviewer #1: No

Reviewer #2: No

---

## [Author Response · Author response to Decision Letter 0]

2 Mar 2022

Comments to Reviewer 1

Thank you for evaluating our manuscript and providing detailed feedback.

"This paper's contribution is international comparison and generalizability with multiple survey sites (France, Germany, Sweden). However, the introduction discussed only findings in a few countries. The survey experiments on COVID-19 vaccine is rapidly increasing and reports finding in various countries (including. for instance, Malawi (Kao, et., al 2021) and Netherlands (Reeskens, et., al. 2021)). I then recommend that the authors update the literature review part, including recent results of survey experiments."

2.1

Thank you for highlighting this additional literature -- research regarding COVID is indeed moving quickly, and we are grateful for the opportunity to include reference to this additional work. Thanks you for bringing these to our attention! As suggested, we added Kao et al. (2021), and Reeskens et al. (2021). We also added some other recent papers such as Argote et al. (2021) for evidence from Latin America; Kawata and Nakabayashi (2021) for evidence from Japan; and Legaspi, Malay and Lim (2021) for evidence from the Philippines (see lines 49, 68-71). 

 

Comments to Reviewer 2

Thank you for evaluating our manuscript and providing detailed feedback.

"Might be worth looking at a public report conducted by the Media Ecosystem Observatory in Canada. They conduct a conjoint experiment, much like Motta (2021) and the authors, that looks at sub-group effects in vaccine attributes in Canada. They find similar results: people who indicate the will take a COVID-19 vaccine are more responsive to these characteristics than those who report they will not or who are uncertain https://www.mcgill.ca/maxbellschool/files/maxbellschool/meo_vaccine_hesistancy_1.pdf"

3.1

Many thanks to the reviewer for this recommendation. We added discussion of this report to our manuscript (see e.g., lines 68–71). 

"A couple other studies provide nice support for the authors’ findings using real-world vaccines. A manuscript by Merkley and Loewen (https://osf.io/ng9qh/) find that people are more likely to differentiate by COVID-19 vaccine brand – as in, reluctance to take AstraZeneca and Johnson & Johnson compared to Pfizer and Moderna – if they are supportive of vaccines, and another article by these authors (JAMA Network Open, 2021) shows people are persuaded to take less preferred vaccines (like AZ and JJ) by highlighting their capacity to prevent death"

3.2

We agree that these studies provide nice support for our findings and have added the references to the discussion section of our manuscript (see lines 555-560). 

"Why France, Germany and Sweden? I’d like to see some justification of the case selection. It might also be useful to provide a brief rundown of the vaccine rollout differences across these cases in terms of timeline and vaccine availability. I encourage the authors to at least include the country-by-country MMs in the main text for readers. I think it’s worth highlighting the unique cross-national scope of this study that gets glossed over by only presenting pooled results. If the results are similar, that’s interesting!"

3.3

We agree with the reviewer that it is important to clarify our case selection strategy, and have highlighted the reasoning and inferential benefits of our case selection (see lines 188–196). We also provide some information on the vaccine rollout, and added country marginal means (MMs) to the main text (see lines 389–403). We think that providing this contextual and country-specific information improves our manuscript. 

"I wonder about the choice of attributes for side effects and effectiveness. Do people have a common understanding of the former? Or are some people thinking “fever” while others are thinking VITT? I recognize other studies have taken a similar approach, but the comparative weakness of the effect for side effects may be due to this lack of precision (and possibly R’s numeracy). As we saw with the VITT fiasco, people really do seem to be responsive to side effect issues. Similarly with effectiveness there is no clarity for respondents if the meaning is “symptomatic and asymptomatic infection” or just symptomatic infection or hospitalization or death. The Merkley and Loewen (2021) piece suggests people are responsive to symptomatic infection prevention and death prevention characteristics. Respondents may not be entirely clear on what is meant by effectiveness. Obviously, the authors cannot re-run these experiments, but I think it’s worth pointing out these limitations and proposing new research unpacks the side effect and effectiveness characteristics."

3.4

Many thanks to the reviewer for pointing this out. We did not provide a definition for side effects or effectiveness in the survey, and we did not ask what respondents’ understandings of “side effects” / “effectiveness” were. As you correctly point out, understandings of side effects and effectiveness might vary, which might, in turn, lead to different responses. In designing the study, we considered a number of different ways to try and present this information that either further explained these concepts, or would improve numerical understanding. We felt that all of our attempts were left wanting in some way. We chose the formulation we did because it reasonably approximates how these concepts are discussed in COVID news coverage (“the vaccine is xx% effective” or “1 in xxx,xxx of patients had a serious side effect”). In this regard, we think that our attributes and levels correspond to the real world information environment people are likely to have when evaluating vaccines. We fully agree that with the reviewer (and by extension Merkley and Loewen) that how these concepts are presented are likely to have important consequences that we are unable to observe. We now acknowledge this limitation in our discussion, cite Merkley and Loewen (2021), and emphasize the importance of considering these potential limitations in future research (see lines 526–530).

"I’d like to see a picture of the conjoint task in the main text"

3.5

We thank the reviewer for this excellent suggestion. We now provide a picture of the conjoint task in the main text (see Fig. 1).

"I’d like to see some more effort to highlight effect sizes. These differences we observe in the means, how sizable are they? Might be useful to express differences in standard deviations and compare them to the sizes that have been found in previous work"

3.6

We appreciate the reviewer’s suggestion to give more attention to our effect sizes. In line with your comment, we now detail in the results section using the AMCEs (see lines 380, 389-40, 424) and compare them to other work in the conclusion (see lines 499–501). Overall, we find that our effect sizes are comparable with what other researchers found.

"These findings, in addition to the works pointed to above, lead to some interesting questions that the authors only start to hint at in their discussion. What are the mechanisms behind these sub-group differences? And what exactly do we make of the responsiveness of vaccine supporters to these vaccine characteristics? Would we really see slippage for vaccines with effectiveness in the 50 percent range in the real world? Or do we only really observe this because respondents are aware there are other available vaccines with superior characteristics? I think it’s worth thinking about whether these sub-group effects would generalize to other areas with less vaccine choice."

3.7

We agree with the reviewer that our discussion is not very specific regarding the psychological mechanisms and practical relevance behind our sub-group differences. As such, we now discuss our sub-group effects in more detail (see lines 510–525).

---

## [Decision Letter · Decision Letter 1]

14 Mar 2022

Which vaccine attributes foster vaccine uptake? A cross-country conjoint experiment

PONE-D-21-35451R1

Dear Dr. Barnfield,

We’re pleased to inform you that your manuscript has been judged scientifically suitable for publication and will be formally accepted for publication once it meets all outstanding technical requirements.

Kind regards,

Jean-François Daoust

Academic Editor

PLOS ONE

Additional Editor Comments (optional):

Reviewers' comments:

Reviewer's Responses to Questions

**Comments to the Author**

1. If the authors have adequately addressed your comments raised in a previous round of review and you feel that this manuscript is now acceptable for publication, you may indicate that here to bypass the “Comments to the Author” section, enter your conflict of interest statement in the “Confidential to Editor” section, and submit your "Accept" recommendation.

Reviewer #2: (No Response)

2. Is the manuscript technically sound, and do the data support the conclusions?

Reviewer #2: Yes

3. Has the statistical analysis been performed appropriately and rigorously? 

Reviewer #2: Yes

4. Have the authors made all data underlying the findings in their manuscript fully available?

Reviewer #2: Yes

5. Is the manuscript presented in an intelligible fashion and written in standard English?

Reviewer #2: Yes

6. Review Comments to the Author

Reviewer #2: The authors have done a great job addressing my relatively minor concerns. I recommend publication.

One small point to build on their discussion on line 561. Not only does that study build on the ecological validity of the authors’ study by showing people have a relative preference for Pfizer and Moderna over AZ and JJ, but that relative preference is much stronger for people who are supportive of vaccines. The vaccine hesitancy don’t really a distinction. This study has now been published at Vaccine (https://www.sciencedirect.com/science/article/pii/S0264410X22001682?via%3Dihub)

7. PLOS authors have the option to publish the peer review history of their article (what does this mean?). If published, this will include your full peer review and any attached files.

Reviewer #2: No

---

## [Editor Report · Acceptance letter]

1 Apr 2022

PONE-D-21-35451R1 

Which vaccine attributes foster vaccine uptake? A cross-country conjoint experiment 

Dear Dr. Barnfield:

I'm pleased to inform you that your manuscript has been deemed suitable for publication in PLOS ONE. Congratulations! Your manuscript is now with our production department. 

Kind regards, 

on behalf of

Dr. Jean-François Daoust 

Academic Editor

PLOS ONE